



# 1  A GIS-based multivariate approach to identify flood
# 2  damage affecting factors

3  Barbara Blumenthal [1,3], Jan Haas [2,3], Jan-Olov Andersson [2]

[1] Risk- and environmental studies, Karlstad University, Sweden

[2] Geomatics, Karlstad University, Sweden

[3] Centre for Climate and Safety, Karlstad University, Sweden

Abstract

This paper investigates causal factors leading to pluvial flood damages, beside rainfall amount
and intensity, in two Swedish cities. Observed flood damage data from a Swedish insurance
database, collected under 13 years, and a set of spatial data, describing topography,
demography, land cover and building type were analyzed through principal component
analysis (PCA). The topographic wetness index (TWI) is the only investigated variable that
indicates a significant relationship with to the number and amount of insurance damage. The
Pearson correlation coefficient is 0.68 for the number of insurance damages and 0.63 for
amount of insurance damages. With a linear regression model TWI explained 41 % of the
variance of the number of insurance flood damages and 34 % of variance of amount of
insurance flood damage.
Future studies on this topic should consider implementing TWI as a potential measure in
urban flood risk analyses.

1.  Introduction

Intense rainfall events are common phenomena in Sweden during the summer months
(Gustafsson et al., 2010;Devasthale and Norin, 2014) and have caused considerable amounts
of economic damage as a result of flooding and disruptions of infrastructures (MSB,
2018;Johansson and Blumenthal, 2009). Blumenthal and Nyberg (2018) concluded that
rainfall intensity during the summer months in Sweden and the occurrence of insurance
damages per day caused by floods were highly correlated and that damage is non-linear rising
with increasing rainfall intensity. The conditions may become worse as frequency and
intensity of extreme rainfalls during the summer months are expected to increase in
Scandinavia as a consequence of climate change (Nikulin et al., 2011;Olsson and Foster,
2014).

A central issue of flood risk management is the analysis and the prediction of flood damages.
In recent years, a major part of the research on these topics has focused on the analysis and
the modelling of riverine floods in large catchments (Merz et al., 2010;Jongman et al., 2012),
while little attention has been paid to floods and flood damage as a result of local intense
rainfall. Traditionally, existing depth-damage models have been adapted and combined with



simulated flood depths, estimated through hydraulic modelling (Van Ootegem et al.,
2018;Spekkers et al., 2013). Only a few studies have used insurance flood damage data as a
proxy for flood damage and/or explorative statistical methods to analyze the influence of
topographic and socio-economic factors on observed flood damage caused by intense rainfall.
Spekkers et al. (2014) investigated the influence of a number of socio-economic, building-
related and topographic variables on rainfall induced insurance damage. The results have
shown that the maximum hourly rainfall intensity, the value of the building, the ground floor
area and the household's income are related to insurance damage, while the slope of the
surrounding terrain was not found correlating. Van Ootegem et al. (2018) compared two
multivariate flood damage models in a study on Belgian pluvial flood events. One model was
based on flood depth and the other one based on rainfall accumulation. For both models, a
number of additional variables could be identified that improved explanatory power. The
authors found that risk awareness had reduced flood damage and that a high income had
reduced building damage, while increasing content damage. Further, topography had an
impact on flood damage, i.e. buildings with a higher location than the surrounding houses
were less damaged.

Kalantari et al. (2014) analysed in a case study road damages after an intense rainfall in the
municipality of Hagfors in western Sweden in August 2004. In nine smaller catchments, the
authors investigated the relationships between road damages and geographical characteristics
such as topography, soil type and land use. The results showed that the specific location's
capability to accumulate water (called TWI – topographical wetness index), road density and
soil properties in the catchment and the local channel slope where related to flood damage of
roads.

Sörensen and Mobini (2017) used precipitation and insurance data from the city of Malmö
(Sweden) to analyse the mechanism leading to floods. Their findings emphasise that flood
damage, apart from rainfall intensity and the distance to main sewer systems, is affected by
topographic factors. The authors pointed out that flood damages are more common in flat
areas and towards and along old watercourses. Torgersen et al. (2017) used a multivariate
approach to identify and rank terrain parameters contributing to urban flooding in the city of
Fredrikstad in south-west Norway with help of insurance damage data. The authors found that
sealed areas upstream of the damaged property are related to insurance flood damage.
Furthermore, the study highlighted that flood damages tend to occur in areas with a concave
curvature, while buildings located in steep slopes are less affected.

Jalayer et al. (2014) discussed the use of the TWI to identify urban flooding risk hotspots in
the city of Addis Ababa and found it useful for the determination of flood-prone sites. In a
study of the city Inverloch, Australia, Pourali et al. (2016) found that TWI is usable for the
identification of areas which have a high risk of flooding by intense rainfall. The authors
suggested the usage of the TWI in land use planning as a first step and a cost-effective
alternative to classic hydraulic modelling.

Kaźmierczak and Cavan (2011) studied social factors connected to flood vulnerability in
Manchester based on flood risk maps. Using principal component analysis, the authors found
that low-income and a high percentage of children and elderly people in the population were
related to increased flood vulnerability.



Overall, these studies indicate the existence of a range of damage-influencing factors. These factors can be summarized as topographic variables, building-related variables, land use-related variables and variables related to the socioeconomic status. The present paper aims to study causal factors leading to pluvial flood damages, beside rainfall amount and intensity, in two Swedish cities. Observed flood damage data from a Swedish insurance database, collected under 13 years, and a set of spatial data, describing topography, demography, land cover and building type were analyzed through principal component analysis (PCA).

## 2. Data and methods

This study covers urban and suburban areas in the cities of Gothenburg and Malmö, in southern Sweden (fig. 1).

### 2.1. Spatial scale and time scale

The study areas were delimited by parishes within a 5 km radius around two rain gauges in the cities of Malmö and Gothenburg. In Malmö, there are two large parishes while the Gothenburg study area consists of eleven smaller parishes. Parish sizes vary from 0.4 to 39.4 km$^2$. The total size of the Malmö study area is with 76.7 km$^2$ larger than the one of Gothenburg (66.8 km$^2$). The choice of areas around rain gauges provides the opportunity for a comparison of rainfall characteristics in the two study areas. The gauges are operated by the Swedish Meteorological and Hydrological Institute (SMHI) and have a temporal resolution of 15 minutes.

The study covers a period of 13 years, from 2001 to 2013. Intense rainfall occurs frequently during the summer months in Sweden, giving impetus to limiting the study to the months of June, July and August.

### 2.2. Insurance damage data

The flood damage records used in this study were obtained from the Swedish Länsförsäkringar insurance group. Länsförsäkringar have a market share of around 35 % on the home insurance market. In Sweden, flood insurance is a basic part of the home insurance without any limitations and the insurance coverage is close to 100 % (Grahn and Nyberg, 2017). The damage data from Länsförsäkringar is appropriate as a proxy for all occurred flood damages. The explicit flood risk of a home or estate does not matter for the price of an insurance policy.

The dataset consists of the flood damage occurrence date, the type of damage (building, estate, home property) and the amount of compensation. Most of the insured properties are homes, houses, home property and private estates. Due to privacy issues, the data we received from the insurance company included, the parish where the damage had occurred instead of the exact geographical positions or addresses.



## 2.3. Flood damage variables

The number of insurance damages and the total amount of insurance compensation (in SEK) were counted from the insurance data on a daily basis. No distinctions could be made for different types of damaged objects (buildings, home property, shops), because that would result in too small sample sizes for statistical analyses. The two flood damage variables – *the number of insurance damage* and *the amount of insurance compensation* were adjusted for the insurance company's market share and parish-wise normalized for the total number of households in the parish. The normalized number of insurance damage is throughout this paper called *NIDnorm* and the normalized amount of insurance compensation *AICnorm*.

## 2.4. Geodata

The geospatial data that were used in the analysis were provided by Lantmäteriet (The Swedish Mapping, Cadastral and Land Registration Authority), Statistiska centralbyrån (SCB, Statistics Sweden), Naturvårdsverket (The Swedish Environmental Protection Agency), Svenska Kyrkan (The Church of Sweden) and SMHI (The Swedish Meteorological and Hydrological Institute).

The geodata listed in Table 1, except data from the Church of Sweden and SMHI, were aggregated and clipped to the study area (parishes), where the statistics were gathered at parish level. The generation of statistics that could not directly be obtained from the original source, but required data processing, is described below.

The 25 land cover classes that are present in the study area and how they were aggregated regarding the amount of urban green space and sealed surfaces are listed in Appendix 2. In Appendix 1 the original names of the used building categories are listed.

*Slope in %*

Slope was calculated for the whole Digital Elevation Model (DEM) before clipping to parish level in order to ensure that correct boundary values were derived. The *Slope* function in the *Spatial Analyst* toolbox in *ESRI ArcMap 10.5.1* was used to generate the slope map. Medium slope values per parish where calculated as an indicator of terrain complexity.

*Topographic Wetness Index (TWI)*

The topographic wetness index represents a specific location's capability to accumulate water. The first algorithm (Eq. 1) was developed by Beven and Kirkby (1979) and calculates

$$TWI = \ln(a/\tan \beta) \qquad (Eq. 1)$$

where $a$ is the upslope area per unit contour length and $\beta$ is the local slope.

In this paper, the SAGA Wetness Index (SWI or $WI_S$) was calculated with the SAGA (System for Automated Geoscientific Analyses (www.saga-gis.org)) freeware. This index uses an iterative method that is suitable for plain areas. A high TWI value means a high capability for





1 water accumulation. The medium TWI values per parish where calculated as an indicator of the
2 capability to accumulate water (Fig. 2a and 2b).

*Surface sealing*

The share of surface sealing was determined through classification of land cover data into
three categories, i.e. surface sealing of 0, 50 and 100 %, respectively. Dominant land cover
classes for surface sealing of 0 % are all urban green and blue spaces such as lawns, forest,
water bodies and watercourses. Dense urban areas, infrastructure, industrial and commercial
areas were treated as 100 % sealed. Land cover and land use classes that contain rather equal
shares of sealed and unsealed surfaces are related to sparsely populated neighborhoods with a
larger share of urban green spaces, camping grounds and outdoor sports facilities. Area (in
12 km$^2$) and share of the three classes per parish was calculated.

3.  Statistical analysis

Due to the fact that no rain statistics are available at parish scale, rainfall intensity could not
be a part of the PCA. Variables describing rainfall intensity in the two cities show relatively
large similarities (Table 3). The number of days with observed rainfall was 566 in Malmö and
585 in Gothenburg, and the average rainfall amount per summer in Malmö was 216 mm and
283 mm in Gothenburg. The *maximum, mean and standard deviation for daily rainfall*
*amount (Rday), maximal hourly precipitation amount (RMAX60min)* and *maximal 15 min*
*precipitation amount (RMAX15min)* were found to be fairly similar for the two cities.

As the next step, Principal Component Analysis (PCA) was performed to investigate the
general relationships between the variables of flood damage, demography, topography, land
cover, residential building type and purchasing power. The variables and variable names are
listed in Table 3. Following the PCA, linear regression (LR) was used to study the covariance
between flood damage variables and TWI. All calculations were carried out with the statistics
software *SPSS 24*.

4.  Results

4.1. Topographic and damage variables

The most important finding is that TWI and the number and the amount of damage (NIDnorm
and AICnorm) are correlated (Fig. 3.). The Pearson correlation coefficient between TWI and
the damage variables is 0.68 for the number and 0.63 for the amount; p<0.001 for both. No
correlation could be found between the topographic variables slope, maximal elevation and
TWI.

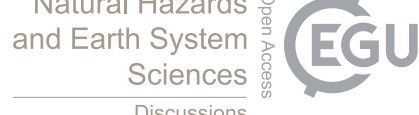

### 4.2. Socio-economic variables

There were strong positive significant correlations within and between the age classes and the number of persons per household (r-value 0.7 to 0.9). Lower, and partly insignificant correlations were found for the age class 20-24. In the first principal component, the age classes 0-6, 7-15, 16-19, 25-44, 45-64 and 65+ were dominating (Fig. 4a), which have been extracted as *a new variable, Extr_Family.*

There are significant correlations in the dataset between the different socio-economic variables, e.g. between family (Extr_Family) and persons per household (Pers_household) or between low purchasing power (Pp_low) and young adults (Age_20to24) (Fig. 4a and 4b). As Table 4 shows, no relationships were found to the flood damage variables. The PCA showed that the third component is solely related to the damage variables (Table 4).

### 4.3. Building type and land cover variables.

The PCA revealed that high ratio of sealing, high purchasing power and high percentage of multi-storey dwellings are correlated with each other, but that they are not related to flood damage. A high percentage of green space (Sealing0), a high number of persons per household, and high percentage of residential areas of row houses and single-family houses are positively correlated to each other, but also here, no relationship with the flood damage variable could be identified. Solely the topographical wetness index (TWI) and the flood damage variables have loadings in the second component, and are significantly positively correlated. The Pearson correlation coefficient is 0.68 for NIDnorm and 0.63 for AICnorm.

Figure 5 displays the component plot of land cover, building type, socio-economic and flood damage variables. The principal component loadings for land cover, building type, socio-economic and flood damage variables are listed in Table 5. The building type variables Villa and Linked house and Apartment house and Row house had strong positive significant correlation; therefore two new variables have been extracted: *Extr_Villa* (Villa and Linked house) and *Extr_row_apart* (Apartment house and Row house).

### 4.4. Linear regression

The results of the principal component analysis identified TWI as the only variable which is linked to the flood damage variables NIDnorm and AICnorm, the Pearson correlation coefficient being 0.68 and 0.63 respectively. Furthermore, a scatterplot of TWI against NIDnorm and AICnorm suggests a linear relation (Fig. 6a and 6b). Hence, a linear regression model was fitted with TWI as independent variable. The model estimations are significant and the model residuals were tested for heteroscedasticity. The results in Table 6 reveals that 41 % ($r_{(adjusted)}$=0.41) of the variance of NIDnorm and 34 % ($r_{(adjusted)}$=0.34) of variance of AICnorm can be explained by the TWI.



## 5. Discussion and conclusions

As stated in the *Introduction*, our initial objective of the study was to identify whether and
how flood damages are related to topography, demography, land cover and building type.
From the PCA it was deduced that TWI is the only investigated variable that indicates a
significant relationship with to the number and amount of insurance damage. The Pearson
correlation coefficient is 0.68 for the number of insurance damages and 0.63 for amount of
insurance damages. With a linear regression model TWI explained 41 % of the variance of the
number of insurance flood damages and 34 % of variance of amount of insurance flood
damage.

The results in this study correspond to a number of previous studies where topographic
characteristics have been investigated and discussed as a contributing factor to flood damages.
Kalantari et al. (2014) investigated a flash flood event in Hagfors (Sweden) and found that
TWI and slope was related to road damages caused by flooding, and Sörensen and Mobini
(2017) emphasized that locations in flat areas and along old watercourses are related to
insurance flood damage in the city of Malmö. Van Ootegem et al. (2018) reported less pluvial
flood damages for buildings located higher than the buildings in the neighborhood in Flanders
(Belgium). Torgersen et al. (2017) highlighted that locations with upstream sealed areas and
locations with a concave curvature are related to flood damage. Jalayer et al. (2014) and
Pourali et al. (2016) suggested the calculating and mapping of the TWI as a first step for
urban flood risk assessment. In contrast to these studies, Spekkers et al. (2014) could not
identify any relation between insurance flood damage and slope in the Netherlands.

Contrary to the expectations and the findings of some previous studies (Jalayer et al.,
2014;Spekkers et al., 2014), this study was unable to demonstrate that building type, degree of
surface sealing and socio-economic factors have an impact on insurance flood damage. A
possible explanation for these results may be the spatial resolution at parish scale. It is,
however, important to notice that the PCA showed the expected spatial relations between the
other investigated variables, i.e. building type, degree of surface sealing and socio-economic
conditions. On one hand, a high degree of sealing, high purchasing power and multi-storey
dwellings are spatially correlated – representing urban city areas – and on the other hand, a
high average number of persons per household, residential areas of row houses and single-
family houses and a high degree of green space are correlated, representing suburban areas
with children families. This indicates that the approach and socioeconomic data used in this
study is suitable for the investigation of spatial relationships of the examined variables.

One limitation of this study is its low spatial resolution (at parish scale). In order to protect the
policyholders' privacies and because of commercial confidentialities, we received the data
from the insurance company in a spatially aggregated form without the exact geographical
position or address of the damaged property. This circumstance led to a relatively small
sample size.



A more refined estimation and classification of percentage of sealed surfaces and retention
areas could reveal new insights into how these factors influence surface run-off and
infiltration patterns and thus resulting in flooded areas. This study relies on official land
use/land cover data from the year 2000. Until today, there is no more recent official land
cover product available for the study area. However, the national dataset is presently updated
by the Lantmäteriet and more actual data is soon to be expected. Integration and analysis of
remote sensing data recorded before the occurrence of a flooding event is recommended in
further studies for a more detailed estimation of sealed surfaces and urban green and blue
spaces. Regarding the role of the socioeconomic factors, further work needs to be done with a
finer spatial resolution of the damage data to establish whether there exists a relationship to
pluvial flood damage or not.

In general, the results of our study highlights the importance of geographic information for
identifying of flood-influencing factors. TWI seems to be a very relevant variable in
explaining urban flood damage. Insurance flood damage data as a proxy for flood damage are
a key for further understanding of the causes and mechanism of pluvial flood damage. The
approach of our study could be applied to identify and more accurately predict pluvial flood
risk in the future. The method could further prove as a time and resource efficient alternative
to traditional depth – damage models and hydraulic modelling. Future studies on this topic
should consider implementing TWI as a potential measure in urban flood risk analyses.



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





1   *Table 1. Geodata used in the analysis.*

| Publisher | Data | Format | Original data product name |
|---|---|---|---|
| Lantmäteriet | Elevation | raster (2 m resolution) | Hojddata2mRaster |
| Lantmäteriet | Real estate | vector (polygon) | FastighetskartanBebyggelseVektor |
| Naturvårdsverket | Land cover | vector (polygon) | Svenska Marktäckedata |
| SCB | Population | vector (polygon) | BefolkningVektor |
| SCB | Purchasing power | vector (polygon) | InkomsterVektor |
| Svenska Kyrkan | Parishes | vector (polygon) | Församlingar |
| SMHI | Gauges | vector (point) | Väderstationer |





1    *Table 2. General statistics of rainfall in the studied areas.*

| Malmö | n | Maximum | Mean | Std. Dev. | Average rainfall sum per summer (mm) | Days with observed rainfall |
|---|---|---|---|---|---|---|
| Rday | 1195 | 65.5 | 2.3 | 5.5 | 216.0 | 566 |
| RMAX60min | 1195 | 25.9 | 1.1 | 2.5 | | |
| RMAX15min | 1195 | 16.5 | 0.7 | 1.5 | | |
| | | | | | | |
| Gothenburg | | | | | | |
| Rday | 1196 | 59.7 | 3.1 | 6.5 | 283.0 | 585 |
| RMAX60min | 1196 | 30.0 | 1.3 | 2.8 | | |
| RMAX15min | 1196 | 15.0 | 0.8 | 1.7 | | |





1    *Table 3.Geodata, damage data and variable names used in the PCA.*

| | |
|---|---|
| AVRslope | Average slope in % |
| TWI | Topographic Wetness Index (TWI) |
| Max_Elevation | Maximum elevation in m |
| Min_Elevation | Minimum elevation in m |
| Age | Residents' age distribution in 7 groups (age from 0-6; 7-15; 16-19; 20-24; 25-44; 45-64 and 65+) |
| Pers_household | Persons per household |
| PP | Total number of households distributed in 4 purchasing power categories (low, medium-low, medium-high and high) |
| Multi_Storey | Percentage multi-story dwelling |
| Villa | Percentage villa |
| Linked_house | Percentage linked house |
| Apartment_house | Percentage apartment house |
| Row_house | Percentage row house |
| Sealing100 | Surface sealing (100 %) in $km^2$ and % |
| Sealing50 | Surface sealing (50 %) in $km^2$ and % |
| Sealing0 | Surface sealing (0 %) = urban green spaces in $km^2$ and % |
| NIDnorm | normalized number of insurance damages |
| AICnorm | normalized amount of insurance compensation |



*Table 4. Principal component loadings for the socio-economic and damage variables.*

|  | Component | | |
|---|---|---|---|
|  | 1 | 2 | 3 |
| Extr_fam |  | 0.894 |  |
| Pp_low | 0.975 |  |  |
| Pp_m_low |  | 0.752 |  |
| Pp_m_high | -0.861 |  |  |
| Pp_high | -0.691 |  |  |
| Pers_household |  | 0.859 |  |
| Age_20to24 | 0.964 |  |  |
| NIDnorm |  |  | 0.966 |
| AICnorm |  |  | 0.944 |
| Percentage of variance explained (%) | 47 | 24 | 17 |





1   *Table 5. Principal component loadings for the variables of land cover, building type, socio-*
2   *economic and flood damage variables.*

|  | PC 1 | PC 2 |
|---|---|---|
| Multi_Storey | -0.918 | |
| Extr_row_apart | 0.901 | |
| Sealing100 | -0.896 | |
| Extr_Villa | 0.864 | |
| Sealing0 | 0.842 | |
| Pers_household | 0.830 | |
| Pp_m_high | -0.815 | |
| NIDnorm | | 0.955 |
| AICnorm | | 0.907 |
| TWI | | 0.850 |
| Percentage of variance explained (%) | 53 | 26 |





1  *Table 6. Linear regression models.*

| Dependent | Independent | r² _adjusted | Regression coefficient (standard error) | Intercept (standard error) | n |
|-----------|-------------|--------------|-----------------------------------------|----------------------------|---|
| NIDnorm | TWI | 0.41 | 0.004** (0.001) | -0.10* (0.005) | 13 |
| AICnorm | TWI | 0.34 | 203.3* (75.8) | -599.6* (267.4) | 13 |

3  * significant at 0.05 level
4  ** significant at 0.01 level





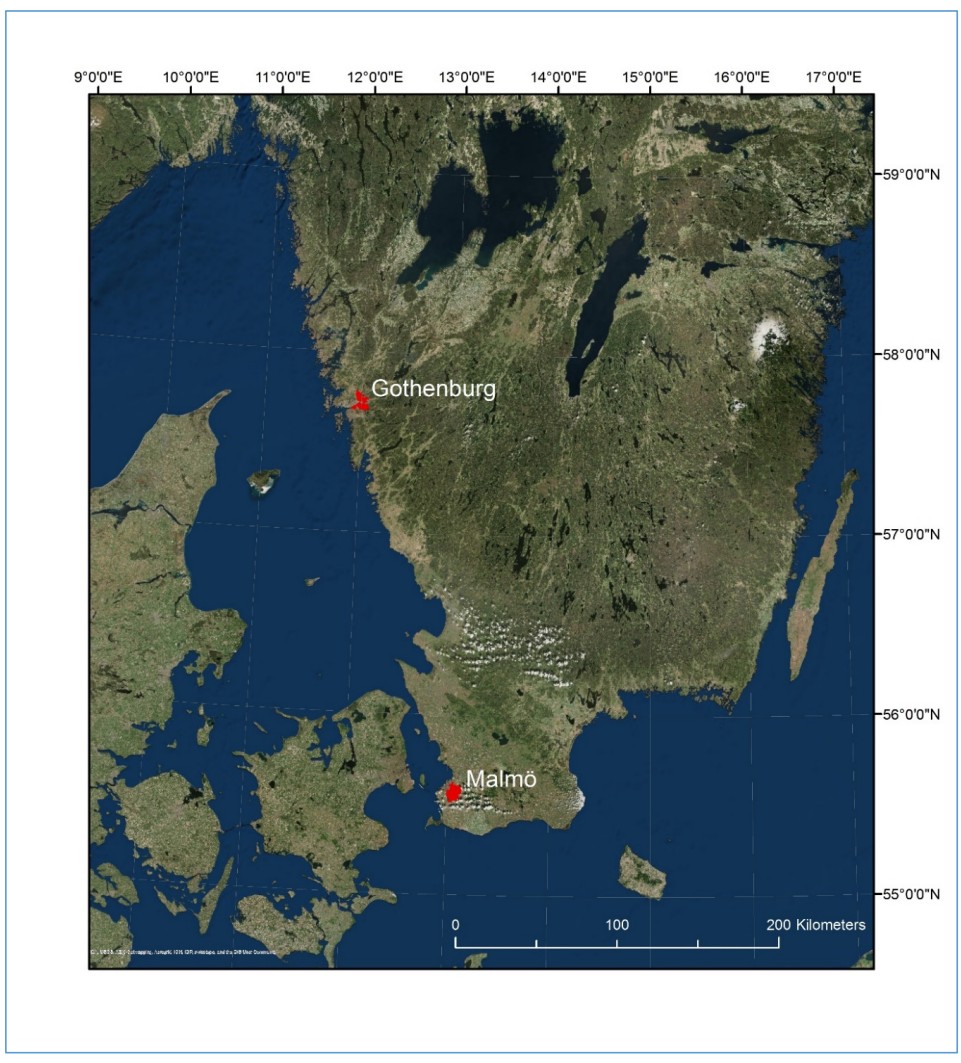

3    *Fig. 1: Location of the study areas in Southern Sweden.*





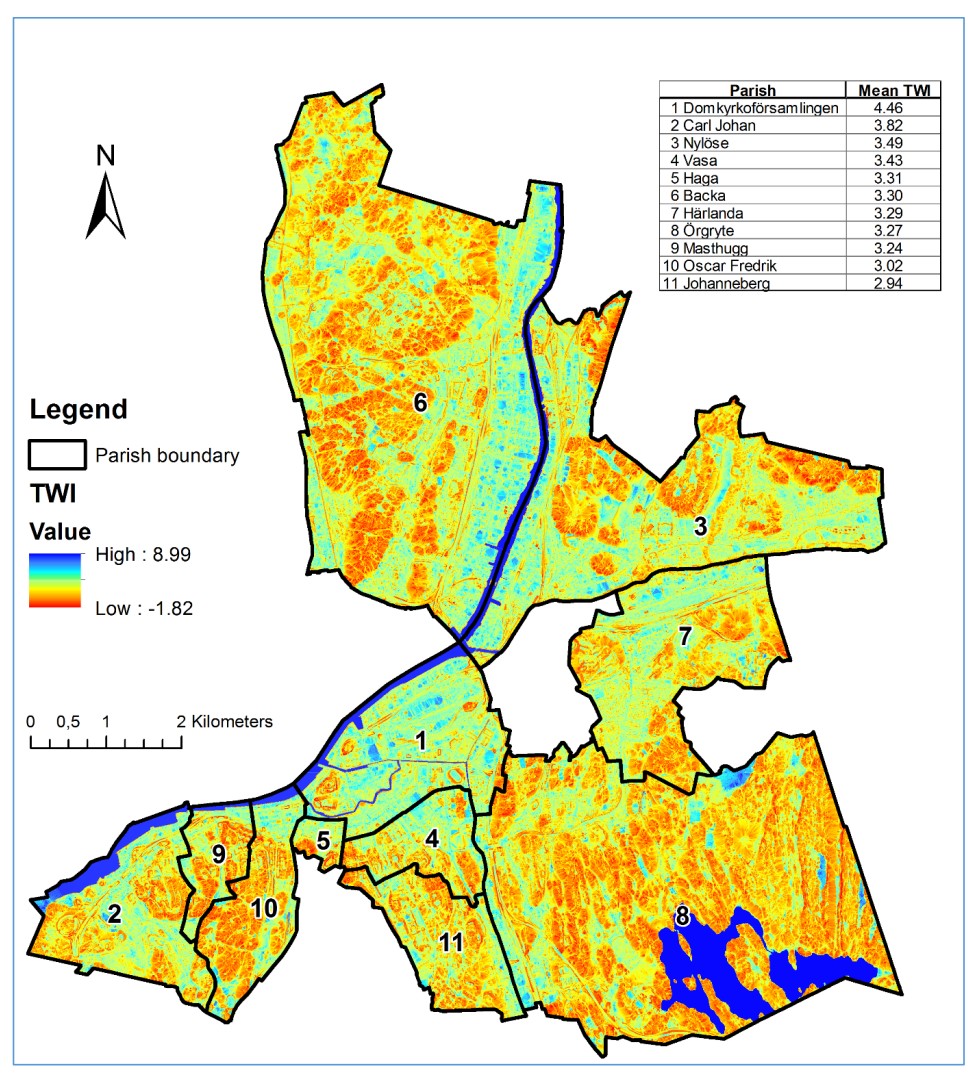

2   *Fig. 2a. Parish boundaries and topographic wetness index (TWI) in Gothenburg.*





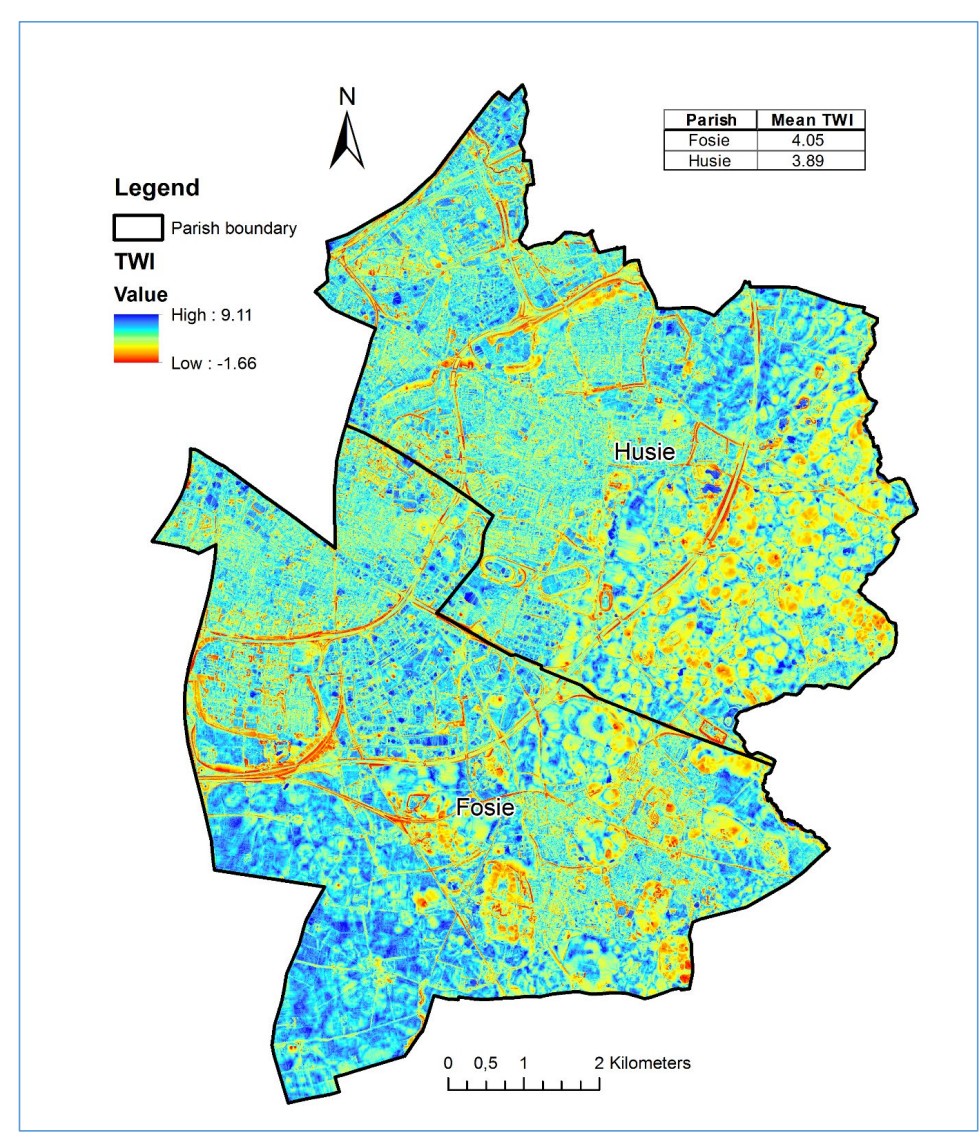

2    *Fig. 2b. Parish boundaries and topographic wetness index (TWI) in Malmö.*





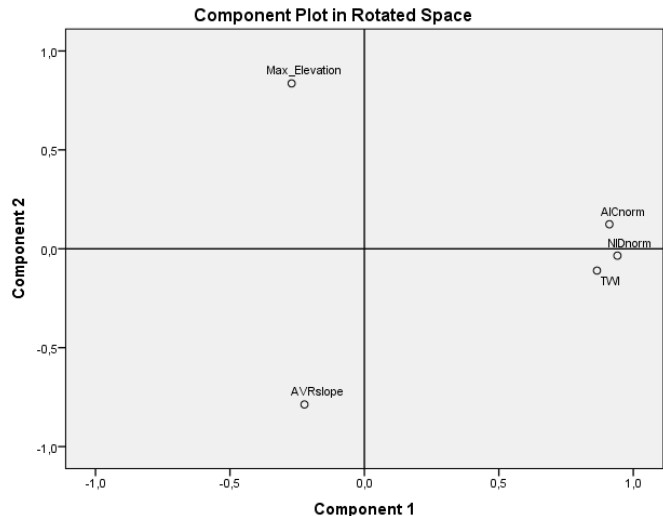

2 *Fig. 3. Component plot for the topographical and damages variables.*



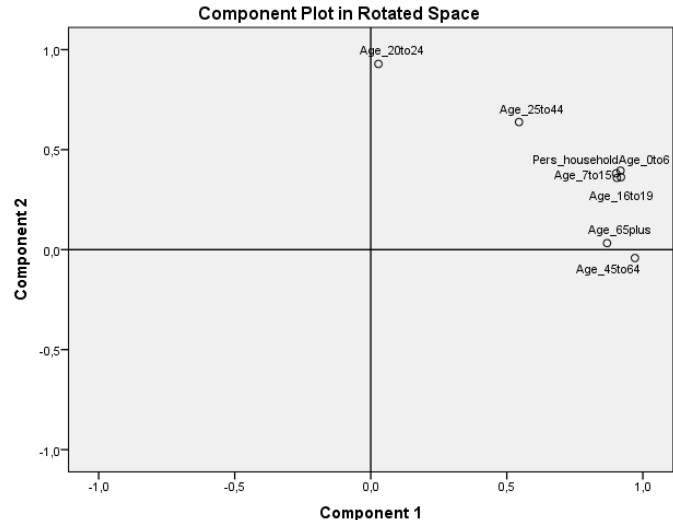

2    *Fig. 4a. Component plot for demographic variables.*



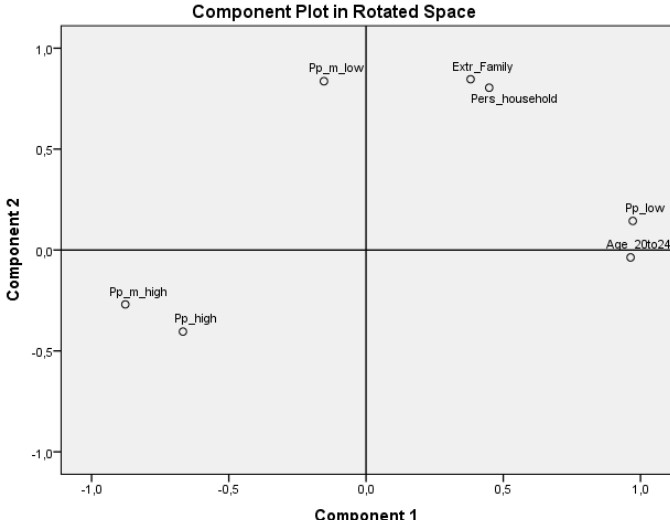

3    *Fig. 4b. Component plot for demographic and economic variables.*





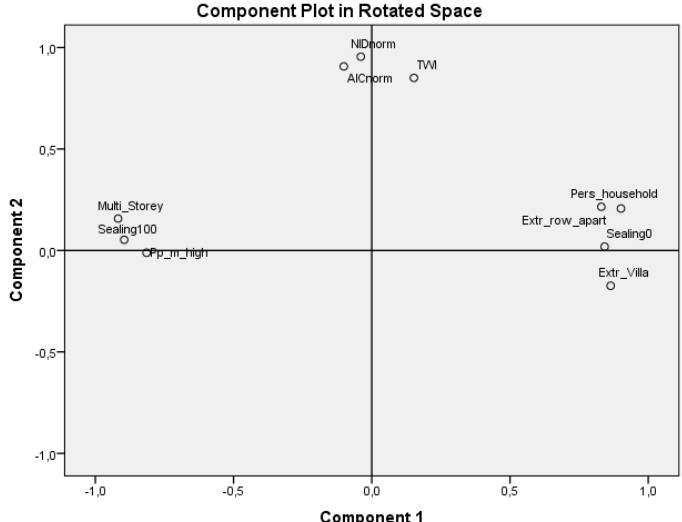

2   *Fig. 5. Component plot of land cover, building type, socio-economic and flood damage*
3   *variables.*



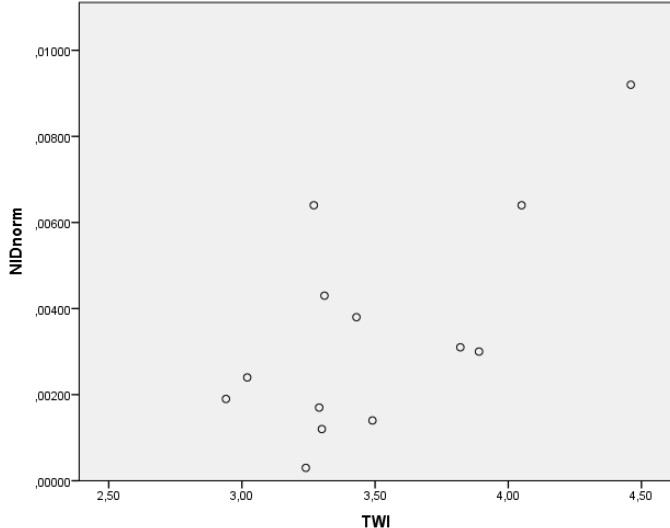

2 *Fig. 6a. Scatterplots illustrating the relationship between TWI and NIDnorm.*



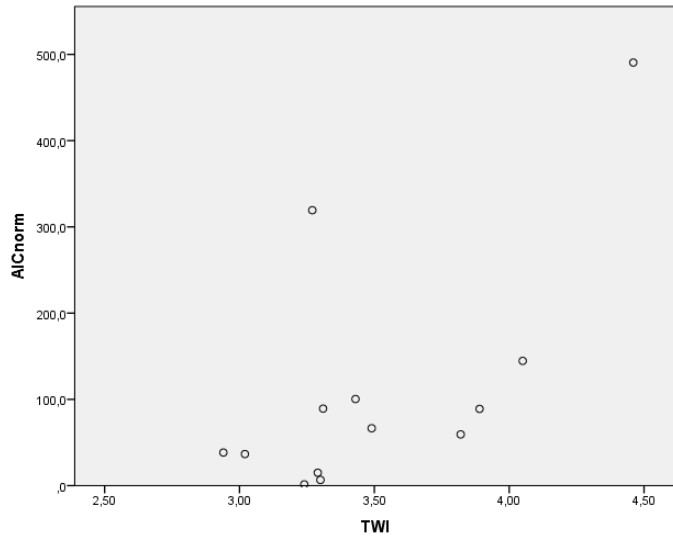

2    *Fig. 6b. Scatterplot illustrating the relationship between TWI and AICnorm.*



1    **Appendix 1**

| Building function/category and specification | Original denotation |
|---|---|
| Residential; multi-story dwelling | Bostad; Flerfamiljshus |
| Residential; villa | Bostad; Småhus friliggande |
| Residential; linked house | Bostad; Småhus kedjehus |
| Residential; apartment house | Bostad; Småhus med flera lägenheter |
| Residential; row-house | Bostad; Småhus radhus |





2    **Appendix 2**

| Land cover class | Original denotation | Surface sealing in % |
|---|---|---|
| Camping grounds | Campingplats och fritidsbebyggelse | 50 |
| Clear-cut area | Hygge | 0 |
| Coniferous forest | Barrskog | 0 |
| Deciduous forest | Lövskog | 0 |
| Dense urban area | Tät stadsstruktur | 100 |
| Estuary | Estuarie | 0 |
| Farmland | Åkermark | 0 |
| Golf course | Golfbana | 0 |
| Industri, commerce, public sector, etc. | Industri, handelsenheter, offentlig service mm. | 100 |
| Marshland | Övrig myr | 0 |
| Mixed forest | Blandskog | 0 |
| Neighbourhood < 200 residents | Orter <200 invånare | 50 |
| Neighbourhood > 200 residents with larger share of green spaces | Orter >200 invånare och med större områden av grönt | 50 |
| Neighbourhood > 200 residents with smaller share of green spaces | Orter >200 invånare och mindre områden av grönt | 100 |
| Non-urban park | Ej urban park | 0 |
| Outcrop | Berg i dagen | 0 |
| Pasture | Betesmark | 0 |
| Road and railroad network | Väg och järnvägsnät med kringområden | 100 |
| Ski slope | Skidpist | 0 |
| Solitary houses and farms | Enstaka hus och gårdsplaner | 0 |
| Sports facilities, shooting range etc. | Idrottsanläggning, skjutbana mm. | 50 |
| Springwood | Ungskog | 0 |
| Urban green spaces | Urbana grönområden | 0 |
| Water bodies | Sjöar och dammar | 0 |
| Watercourses | Vattendrag | 0 |

