# Peer review of "A GIS-based multivariate approach to identify flood 1"

_Natural Hazards and Earth System Sciences, 2018_

## Referee Comment (RC1) · S. Singh (Referee) · 19 Nov 2018

The title of the manuscript A GIS-based multivariate approach to identify flood damage affecting factors is really interesting. The authors discussed causal factors leading to pluvial flood damages, beside rainfall amount and intensity, through PCA and TWI index. The abstract is not clearly written. Which criteria authors have used for classification of slope? Kindly mention. Mention the version of SAGA software. Kindly provide few citations of PCA. The PCA is a black box technique. Have you normalized the different variables, if yes then what is the order of normalization?

---

## Author Comment (AC1) · 22 Nov 2018

Thank you for the time you have invested and for valuable comments which will help us improve the manuscript. Slope was calculated on the whole Digital Elevation Model (DEM) before clipping to parish level in order to ensure that correct boundary values were derived. The Slope function in the Spatial Analyst toolbox in ESRI ArcMap 10.5.1 was used to generate the slope map. The slope function identifies the slope (steepness) from each cell of a raster using a 3x3 moving window to process the data. A more detailed description of slope calculation is found here: http://desktop.arcgis.com/en/arcmap/latest/tools/spatial-analyst-toolbox/slope.htm. Medium slope values per parish where calculated as indicator of terrain complexity. The SAGA Wetness Index was calculated with SAGA 5.0.0. In the

[Figure]

PCA we standardized the variables, used varimax and determined the dimension by Kaiser's eigenvalue > 1 rule in combination with Scree plotting (Backhaus et al., 2013). In a similar context, the procedure is even described in Kalantari et al. (2014) and Kaźmierczak and Cavan (2011), mentioned in our introduction.

Backhaus, K., Erichson, B., Plinke, W., Schuchard-Ficher, C., and Weiber, R.: Multivariate analysemethoden: eine anwendungsorientierte einführung, Springer-Verlag, 2013. Kalantari, Z., Nickman, A., Lyon, S. W., Olofsson, B., and Folkeson, L.: A method for mapping flood hazard along roads, Journal of Environmental Management, 133, 69-77, 10.1016/j.jenvman.2013.11.032, 2014. Kaźmierczak, A., and Cavan, G.: Surface water flooding risk to urban communities: Analysis of vulnerability, hazard and exposure, Landscape and Urban Planning, 103, 185-197, https://doi.org/10.1016/j.landurbplan.2011.07.008, 2011.

---

## Editor Comment (EC1) · Patel (Editor) · 31 Dec 2018

Paper is relevant to the scope of the journal, it describes the importance of TWI for urban flood risk analysis. The topic is certainly in interest of the international reader; However, I found several concerns that deserved to be addressed to improve the quality of the paper. Major comments: 1) Pg.4, line 25, Provide the details of DEM which used for derive the slope map of the study area. 2) Fig.1 North arrow is missing 3) Author(s) has classified the land cover in three classes, however the description of land use classification methods and source image is missing 4) Author(s) have pertained the average rainfall in Table 2, however the method used for calculation of Average rainfall is missing. In addition, study is based on spatially located data bases although the location map of rain gauages are missing. 5) Pg. 3, Line 21 temporal resolution is

15 minutes, however as per table 2 temporal resolution has been considered for 15 minutes and 60 minutes, it shows the wrong interpretation of data sets for statistics. 6) Pg. 3, line 39 shows that Geographic positions for damages are not certain then how it is compare with TWI analysis? 7) What is PCA? What its significance flood damage analysis? 8) Pg. 6, line 3, what is the r value of 0.7 to 0.9? Explain it significant in present study Minor Comments: 1) Pg. 7, line 3, The word introduction in italic font, use it in normal text 2) It has observed that author(s) has used Swedish name at many places; kindly provide the English name for the understanding of wide range of international reader. 3) Pg. 4, line 22 and 23, Appendix 2 cited before Appendix 1 in text. 4) Many sentence structure formations are wrong, i.e. page 5, line 1-2, The medium TWI….., which is not clarify its meaning. Many grammatical mistakes and sentence structure formation errors are observed throughout the manuscript. It is important to verify it with native English speakers. Please discuss why this study is useful to the local communities, and how they could use the information.

---

## Referee Comment (RC2) · Anonymous Referee #2 · 6 Jan 2019

The paper introduces a GIS-based multivariate approach 'to identify flood damage factors'. It uses two Swedish cities as case-study locations, concluding that the topographic wetness index (TWI) is the main variable explaining the number and amount of insurance damage for the specific case-study.

The study is essentially a very simple sensitivity analysis, on a specific case-study. In my opinion, the paper doesn't introduce any substantial contribution to the field to be published in a highly-regarded journal such as NHESS. There is some value in the particular damage dataset used. However, the analysis method is very simplistic (at undergraduate thesis level) and all the major literature on the topic of flood damage is ignored, for instance, the work by Heidi Kreibich's research group (many papers in the same journal).

[Figure]

There is a substantial lack of critical discussion on the obtained results; it is also not clear what a reader should take away from the study and what is the usefulness of the study. The main conclusion is that 'Future studies on this topic should consider implementing TWI as a potential measure in urban flood risk analyses'. However, it is very well-known that the use of TWI has several limitations and that probabilistic flood risk models would require much more advanced proxies, methods and tools (hydrological + hydraulic models; exposure information, damage functions; etc.).

Some specific comments:

1) How do the authors discriminate rainfall-induced insurance damage from generic flood insurance damage? I am not convinced that insurance policy/claims have this level of detail.

2) 'The explicit flood risk of a home or estate does not matter for the price of an insurance policy': what do the authors mean here? This concept doesn't seem to make sense. If there is no link between insurance policy (and claims) and flood risk, why then this study is needed, considering that TWI could be seen as a very (very!) rough proxy for flood risk?

3) What is the number of insurance damage? Just the number of assets/claims? I am also not convinced by the specific normalization performed in the study. The common flood risk models simply consider loss ratios (repair vs replacement) as the main 'output' variable to be correlated with some local intensity proxy (water depth/velocity, etc).

4) Is TWI the same of the SWI? Why do the authors use two different definitions?

5) Rainfall intensity is not part of the PCA performed by the authors simply because 'no rain statistics are available at parish scale'. This is not a good justification as the physics of a given phenomenon should always come first, independently of the available data.

6) It is very surprising that other variables than TWI play such a minor role in explaining flood damage. Some critical discussion on this aspect would have been beneficial.

7) Very poor-quality figures; lots of typos and unclear sentences throughout the manuscript.

---

## Author Comment (AC2) · 10 Mar 2019

'Paper is relevant to the scope of the journal, it describes the importance of TWI for urban flood risk analysis. The topic is certainly in interest of the international reader; However, I found several concerns that deserved to be addressed to improve the quality of the paper'

Thank you for the time you have invested and for valuable comments which will help us improve the manuscript. We have considered all comments. The changes are listed below the individual comments. If we for some reason have not been able to accommodate a suggestion an explanation has been provided.

Comment 1: 'Pg.4, line 25, Provide the details of DEM which used for derive the slope

map of the study area.'

Response: Thank you for this comment. DEM details were added to the section. The following was added: "The DEM has 2m horizontal and between 0.05 and 0.2m vertical resolutions and is based on the RH2000 elevation system. The DEM is the official Swedish elevation model (NNH) generated through airborne laserscanning."

Comment 2: 'Fig.1 North arrow is missing' Response: We refrain from using a north arrow because true north varies in the area due to meridian convergence. North is clearly indicated by the latitude/longitude gridnet.

Comment 3: 'Author(s) has classified the land cover in three classes, however the description of land use classification methods and source image is missing.'

Response: Land cover was not classified in the sense of deriving new classes based on spectral/textural information through image classification. Therefore, no confusion matrix is attached either. Official Swedish land cover data that is based on the CORINE classification scheme (Bossard et al. 2000 – CORINE land cover technical guide – Addendum 2000), and that the Swedish Environmental Protection Agency is responsible for, was aggregated from originally 25 classes into 3 classes, each representing a particular percentage of impervious surfaces (class 1 = 0%; class 2 =50% and class 3 = 100%). A clarification of land cover aggregation was provided under "Surface sealing"A class number was assigned to the table in Appendix 2 based on sealed surface percentages. The CORINE reference (Bossard et al. 2000) was added to the reference section. The introductory "Geodata" section was reformulated. The "Surface sealing" section was reformulated.

Comment 4: 'Author(s) have pertained the average rainfall in Table 2, however the method used for calculation of Average rainfall is missing. In addition, study is based on spatially located data bases although the location map of rain gauages are missing.'

Response: The average was calculated as the sum of the total rainfall amounts for

[Figure]

June, July and August from 2001 to 2013 divided with 13. A footnote was added in the manuscript. We added the location of the rain gauges in fig 2a and 2b.

Comment 5: 'Pg. 3, Line 21 temporal resolution is 15 minutes, however as per table 2 temporal resolution has been considered for 15 minutes and 60 minutes, it shows the wrong interpretation of data sets for statistics.'

Response: The finest resolution of the rainfall accumulation is 15 min. We do not understand this comment. If 15 min accumulations are known, even 60 min rainfall accumulations are known..? We have clarified in the manuscript that 15 min is the finest resolution.

Comment 6: 'Pg. 3, line 39 shows that Geographic positions for damages are not certain then how it is compare with TWI analysis?' Response: We calculated the mean TWI of the parishes and analyzed its correlation to the number and the total amount of flood insurance damage per parish.

Comment 7: 'What is PCA? What its significance flood damage analysis?'

Response: PCA (Principal component analyses) is a multivariate method, often used in exploratory data analysis. It is aimed to structure, simplify and visualize a dataset of possibly correlated variables. The original dataset (of correlated variables) converts by orthogonal transformation into a set of uncorrelated variables called principal components (PCs). In this study, PCA is used to analyze which of the investigated variables are correlated to flood damages and which are not. We will develop this issue in the manuscript in the section Statistical analysis.

Comment 8: 'Pg. 6, line 3, what is the r value of 0.7 to 0.9? Explain it significant in present study'

Response: Thank you for the comment. Here we mean the Pearson correlation coefficient r (ranges between -1 and 1). It is a measure for strength and direction of a linear relationship between two variables. R-values between 0.7 and 0.9 indicate a strong

positive relationship. We clarified this point in the manuscript.

Bossard, M., Feranec, J., Otahel, J., 2000. CORINE land cover technical guide: Addendum 2000. Technical report No 40. Copenhagen (EEA).

---

## Author Comment (AC3) · 10 Mar 2019

minor comments:

'Pg. 7, line 3, The word introduction in italic, font, use it in normal text.,

Yes, thank you. Changed to "Introduction section".

'It has observed that author(s) has used Swedish name at many places; kindly provide the English name for the understanding of wide range of international reader.'

Thank you for this comment. Place names such as parishes are not translated since there is no English equivalent. Names of official Swedish authorities are now used throughout the text with their official English denotations rather than the Swedish ones.

[Figure]

The reason why we kept the official Swedish names for land cover and residential building categories in Appendix 1 and 2 is that readers should be given the possibility to trace back original data descriptions and review meta data since there is no official Swedish translation.Emphasis has been placed on using the English name for Swedish Authorities throughout the text and appendices instead of the Swedish ones. Appendix1 and Appendix 2 were reviewed and reworked. The whole manuscript was checked for further Swedish names that needed to be translated.

'Pg. 4, line 22 and 23, Appendix 2 cited before Appendix 1 in text.'

Thank you, our mistake. The text has been reformulated.

'Many sentence structure formations are wrong, i.e. page 5, line 1-2, The medium TWI: : :.., which is not clarify its meaning. Many grammatical mistakes and sentence structure formation errors are observed throughout the manuscript. It is important to verify it with native English speakers. Please discuss why this study is useful to the local communities, and how they could use the information.'

In Sweden, municipalities are solely responsible for all spatial planning processes as well as flood risk management. Therefore, municipalities are interested in methods to identify flood risk due to intense rainfall on local level. The calculation and mapping of the SWI could be the starting point in flood risk assessment related to intense rainfall. We will develop this issue in the Discussion section. The language has been revised throughout the manuscript and spelling errors and formulations were addressed.

---

## Author Comment (AC4) · 10 Mar 2019

Referee #2: 'The paper introduces a GIS-based multivariate approach 'to identify flood damage factors'. It uses two Swedish cities as case-study locations, concluding that the topographic wetness index (TWI) is the main variable explaining the number and amount of insurance damage for the specific case-study. The study is essentially a very simple sensitivity analysis, on a specific case-study. In my opinion, the paper doesn't introduce any substantial contribution to the field to be published in a highly-regarded journal such as NHESS. There is some value in the particular damage dataset used. However, the analysis method is very simplistic (at undergraduate thesis level) and all the major literature on the topic of flood damage is ignored, for instance, the work by Heidi Kreibich's research group (many papers in the same journal). There is a substantial lack of critical discussion on the obtained results; it is also not clear what a reader should take away from the study and what is the usefulness of the study. The main conclusion is that 'Future studies on this topic should consider implementing TWI as a potential measure in urban flood risk analyses'. However, it is very well-known that the use of TWI has several limitations and that probabilistic flood risk models would require much more advanced proxies, methods and tools (hydrological + hydraulic models; exposure information, damage functions; etc.).'

Authors: We thank the reviewer for the time invested and for valuable comments. We have considered all comments and made changes accordingly. If we for some reason have not been able to accommodate a suggestion an explanation has been provided.

The scope of this study is insured flood loss caused by intense rainfall and the literature review in the Introduction section focused on this topic. Heidi Kreibichs group is citied in the original manuscript on P1L36 in (Merz et al., 2010). Please give us more some specific relevant examples on literature on flood damages caused by intense rainfall we have missed to consider in the manuscript. The relative simplicity of the approach is a consequence of the resolution of the damage dataset at parish level. Our study is an explorative analysis of 13 years of empirical flood damage data collected by an insurance company in Sweden's second and third largest cities. It took us almost a year of negotiation to obtain the damage records. We received quite unexpected results, which we want to share with the research community.

We agree, the sentence 'Future studies on this topic should consider implementing TWI as a potential measure in urban flood risk analyses' cannot stand alone in the abstract and needs some more explanation. Our point is to use TWI/SWI as a first step in urban flood risk assessment to identify flood risk hotspots and then go on with more advanced hydrological/hydraulic modeling at those locations. We will rewrite the abstract, especially the alone standing main conclusion. In the Discussion section, we will clarity that TWI/SWI only can be seen as a rough proxy for flood risk and used as an identification tool.

Comment 1: 'How do the authors discriminate rainfall-induced insurance damage from generic flood insurance damage? I am not convinced that insurance policy/claims have this level of detail.'

Response: The data originates from an insurance database that was specifically created for flood damage. The database contains solely information about damages caused by flooding from rivers, streams, rainfall, groundwater or sewer systems. We clarify this in the Insurance data section.

Comment 2: 'The explicit flood risk of a home or estate does not matter for the price of an insurance policy': what do the authors mean here? This concept doesn't seem to make sense. If there is no link between insurance policy (and claims) and flood risk, why then this study is needed, considering that TWI could be seen as a very (very!) rough proxy for flood risk?'

Response: Yes, that is correct. The Swedish insurance system, at least on the home insurance sector, still works solidary. Furthermore, this study is not aimed at helping insurance companies to adjust their premiums, but to identify flood damage affecting factors related to intense rainfall events.

Comment 3: 'What is the number of insurance damage? Just the number of assets/claims? I am also not convinced by the specific normalization performed in the study. The common flood risk models simply consider loss ratios (repair vs replacement) as the main 'output' variable to be correlated with some local intensity proxy (water depth/velocity, etc).'

Response: Exactly, the number of insurance damage is the number of the insurance payouts in the parishes. Loss ratios require the values of buildings and inventories. The insurance company uses very rough templates in this case and they are not convinced themselves regarding how far those templates reflect the real values. Otherwise, property tax assessment value are available to construct loss ratios. These values however are strongly determined by the market value/the location of the property.

Comment 4: 'Is TWI the same of the SWI? Why do the authors use two different definitions?'

Response: Thank you for this comment. We have replaced TWI with SWI throughout the manuscript as it is de facto the SWI that was used. A description of the index and motivation for its use was added to the topographic wetness section alongside two references. "TWI" was replaced by "SWI" throughout the manuscript. The section "Wetness Index" was reworked. 2 references were added to the list (Böhner et al., 2002; Böhner and Selige, 2006).

Comment 5: 'Rainfall intensity is not part of the PCA performed by the authors simply because 'no rain statistics are available at parish scale'. This is not a good justification as the physics of a given phenomenon should always come first, independently of the available data.'

Response: Thank you for this comment. We agree, that sounds cryptic and needs some more explanation. Rainfall intensity is of course the primary cause of flood damage related to intense rainfall events. We analyzed the role of rainfall intensity the in the examined study areas in Malmö and Gothenburg using linear regression methods (Blumenthal and Nyberg, 2018)(in ref list P9L4) and found considerable high degrees of explanation. Insurance loss caused by floods was exponentially or potentially increasing with rainfall intensity. Furthermore, even if rain statistics would have been available at parish scale, rain intensity had not been suitable as a variable in a PCA. Rain intensity cannot be represented by a medium values or amounts as the other variables used in the PCA. We will develop this topic in section Statistical analysis.

Comment 6: 'It is very surprising that other variables than TWI play such a minor role in explaining flood damage. Some critical discussion on this aspect would have been beneficial.'

Response: We agree, these results were much unexpected and we will further investigate/discuss this.

Comment 7: 'Very poor-quality figures; lots of typos and unclear sentences throughout the manuscript.'

Response: We partly agree with this comment regarding language and spelling mistakes. It is however unclear to the authors what is meant by "very poor-quality figures." Since no other specification is given here regarding e.g. cartographic representation, map layout or plot size, the authors assume the comment refers to figure resolution. It is very common that lower-resolution figure are chosen with the first submission of a manuscript. High-quality figures (300 dpi+) will be presented upon acceptance of the article. The language has been revised throughout the manuscript and spelling errors and formulations were addressed.

Böhner, J., Koethe, R., Conrad, O., Gross, J., Ringeler, A., and Selige, T. Soil regionalisation by means of terrain analysis and process parameterisation. In: Micheli, E., Nachtergaele, F., Montanarella, L. (Ed.): Soil Classification 2001. European Soil Bureau, Research Report No. 7, EUR 20398 EN, Luxembourg, 213-222, 2002.

Böhner, J., and Selige, T.: Spatial prediction of soil attributes using terrain analysis and climate regionalisation. Göttinger Geogr. SAGA-Analyses and Modelling Applications. Abh, (115), 2006.